# The HSF1-CPT1a Pathway Is Differentially Regulated in NAFLD Progression

**DOI:** 10.3390/cells11213504

**Published:** 2022-11-04

**Authors:** Wiebke Breternitz, Friedrich Sandkühler, Frauke Grohmann, Jochen Hampe, Mario Brosch, Alexander Herrmann, Clemens Schafmayer, Christian Meinhardt, Stefan Schreiber, Alexander Arlt, Claudia Geismann

**Affiliations:** 1Laboratory of Molecular Gastroenterology & Hepatology, Department of Internal Medicine I, UKSH-Campus Kiel, 24105 Kiel, Germany; 2Medical Department 1, University Hospital Dresden, Dresden University of Technology (TU Dresden), 01307 Dresden, Germany; 3Center for Regenerative Therapies Dresden (CRTD), Dresden University of Technology (TU Dresden), 01307 Dresden, Germany; 4Department of General, Visceral, Vascular and Transplantation Surgery, University of Rostock, 18057 Rostock, Germany; 5Department for Gastroenterology and Hepatology, University Hospital Oldenburg, Klinikum Oldenburg AöR, European Medical School (EMS), 26133 Oldenburg, Germany

**Keywords:** NAFLD, bariatric surgery, DNA-methylation, lipid oxidation

## Abstract

Obesity and obesity-associated diseases represent one of the key health challenges of our time. In this context, aberrant hepatic lipid accumulation is a central pathological aspect of non-alcoholic fatty liver disease (NAFLD). By comparing methylation signatures of liver biopsies before and after bariatric surgery, we recently demonstrated the strong enrichment of differentially methylated heat shock factor 1 (HSF1) binding sites (>400-fold) in the process of liver remodeling, indicating a crucial role of HSF1 in modulating central aspects of NAFLD pathogenesis. Using cellular models of NAFLD, we were able to show that HSF1 is activated during fat accumulation in hepatocytes, mimicking conditions in patients before bariatric surgery. This induction was abolished by starving the cells, mimicking the situation after bariatric surgery. Regarding this connection, carnitine palmitoyltransferase 1 isoform A (CTP1a), a central regulator of lipid beta-oxidation, was identified as a HSF1 target gene by promoter analysis and HSF1 knockdown experiments. Finally, pharmacological activation of HSF1 through celastrol reduced fat accumulation in the cells in a HSF1-dependent manner. In conclusion, we were able to confirm the relevance of HSF1 activity and described a functional HSF1-CPT1a pathway in NAFLD pathogenesis.

## 1. Introduction

Non-alcoholic fatty liver disease (NAFLD) has emerged as the most prevalent chronic liver disease worldwide in recent years. It is defined as hepatic steatosis in the absence of secondary causes of hepatic fat accumulation and is associated with insulin resistance and diabetes. While pure steatosis is a largely benign condition, it can be complicated by non-alcoholic steatohepatitis (NASH), which can progress to cirrhosis and liver failure. Furthermore, many epidemiological studies have established that patients with NAFLD are at a higher risk of developing hepatocellular carcinoma (HCC) [1]. The pathogenesis of NAFLD is multifactorial and triggered by environmental factors such as hypercaloric nutrition, lack of physical activity, and genetic predisposition. NAFLD particularly affects obese individuals, with prevalence rates proportional to the severity of excess fat mass and adipose tissue dysfunction [2]. Fat accumulates in the liver in the form of triglycerides stored in lipid droplets, which develops with increased lipotoxicity from high levels of free fatty acids (FFAs), free cholesterol, and other lipid metabolites. Subsequently, mitochondrial dysfunction with oxidative stress and the production of reactive oxygen species (ROS) and endoplasmic reticulum (ER) stress are activated and lead to hepatic inflammation [3]. Bariatric surgery is the most radical therapy for metabolic syndrome and NAFLD, leading typically to significant weight loss, improvement of liver histology, and overall mortality [2,3,4,5,6,7,8,9]. By using unbiased array-based DNA methylation and mRNA expression profiling approaches, we were able to show strong enrichment of differentially methylated heat shock factor 1 (HSF1) binding sites (>400-fold) in liver tissue of NAFLD patients in the process of liver remodeling after significant weight loss following bariatric surgery [10]. This evolutionarily highly conserved transcription factor plays a central role in cellular homeostasis. Although the main focus has been on HSF1-induced expression of chaperone genes, so-called heat shock proteins, HSF1 controls a wide set of target loci in stressed cells and directs a variety of different processes, also in non-stressed conditions, including development, metabolism, and aging [11]. However, the exact role of HSF1 in NAFLD is controversial, since some reports have shown a beneficial effect of HSF1 activation [12,13,14] and others have proposed a role of HSF1 in the progression of the disease.

This study aimed to investigate the regulation of HSF1, the possible role of the transcription factor, and to decipher possible downstream targets in the pathogenesis of NAFLD, with regard to the development of the disease. We confirmed the expression of HSF1 in hepatocytes in biopsies of patients before and after bariatric surgery. Thus, by using an established cellular model of different NAFLD disease steps, we mimicked the process of liver remodeling following weight loss and possible pharmacological intervention strategies.

## 2. Materials and Methods

### 2.1. Cell Culture

The human hepatoma cell lines HepG2 (RRID:CVCL_0027) and Hep3B (RRID:CVCL_0326) were obtained from Leibniz Institute DSMZ-German Collection of Microorganisms and Cell Cultures GmbH (Braunschweig, Germany), cultured in DMEM high glucose (#P04-03500, PanBiotech, Aidenbach, Germany) supplemented with 10% FCS (#F7524, Sigma Aldrich, Darmstadt, Germany), 2 mM L-glutamine (#P04-80100, PanBiotech), and 1 mM sodium pyruvate (#P04-43100, PanBiotech). Cells were incubated at 37 °C with 5% CO_2_ at 85% humidity. Cell lines were tested for Mycoplasma contamination using the MycoAllert Kit (#LT07-418, Lonza, Basel, Switzerland).

To induce fat accumulation, a combination of long-chain FFAs, namely palmitic acid (PA; # P0500, Sigma-Aldrich 16:0; 0.125 mM) and oleic acid (OA; #O1008, Sigma-Aldrich; 18:1 cis-9; 0.125 mM), was administered to HepG2 and Hep3B cells. Fatty acids were conjugated to fatty acid-free BSA (#126579, Sigma-Aldrich), as described previously [15]. Briefly, PA and OA were dissolved in NaOH 0.1 M to yield a 50 mM final concentration and heated until dissolved (at 90 °C or 70 °C, respectively). Afterward, 50 mM PA and OA solution was added to 10% fatty acid-free BSA and incubated at 37 °C for 15 min. Finally, H_2_O was added to result in a concentration of 5 mM of fatty acids. Media supplemented with fatty acid-free bovine serum albumin (BSA) was used as a control. To activate HSF1, celastrol (#C0869, Sigma-Aldrich) was added to cells in different concentrations (0.025–0.2 mM).

### 2.2. Patient Samples

Liver samples were obtained intraoperatively during bariatric surgery for assessment of liver histology. Biopsies were immediately frozen in liquid nitrogen, ensuring an ex vivo time of less than 40 s in all cases. A percutaneous follow-up biopsy was collected 5–9 months after surgery. Patients with evidence of viral hepatitis, hemochromatosis, or alcohol consumption greater than 20 g/day for women and 30 g/day for men were excluded. All patients provided written informed consent. The study protocol was approved by the institutional review board (‘‘Ethikkommission der Medizinischen Fakultät der Universität Kiel,’’ D425/07, A111/99) before the commencement of the study. Standardized histopathological assessment [16] was performed by a single pathologist blinded to the molecular analyses. Furthermore, the NAFLD Activity Score (NAS) was calculated (https://www.mdcalc.com/calc/10068/nafld-non-alcoholic-fatty-liver-disease-activity-score, accessed on 15 August 2022) and the clinical features are summarized in Table 1.

### 2.3. Immunohistochemistry

For immunohistochemistry, formalin-fixed and paraffin-embedded (FFPE) sections were used. After deparaffinization, tissue sections were pretreated with citrate buffer (pH 6) for antigen retrieval and incubated with hydrogen peroxide block (TA-060-HP, ThermoFischer Scientific, Waltham, MA, USA) and Ultra V Block (ThermoFischer Scientific) to avoid unspecific reactions. Immunostaining was performed using a rabbit polyclonal anti-HSF1 antibody (Sigma-Aldrich Cat# HPA008888, RRID:AB_1079088; 1:500) in a moist chamber at room temperature for 30 min, following incubation overnight at 4 °C. Slides were washed between steps with Tris-buffered saline (TBS). Immunoreactions were visualized with the N-Histofine Simple Stain MAX PO System (#414142F, Nichirei Bioscience, Tokyo, Japan) and DAB substrate (#SK-4100, Vector Laboratories, Burlingame, CA, USA). The specimens were counterstained with hematoxylin (#MHS128, Merck, Darmstadt, Germany). The omission of the primary antibody served as a negative control. For evaluation, sections were assessed using the intensity of staining (S) (0: no staining; +1: weak; +2 moderate, +3 strong) and the average percentage of immunoreactive cells (P) was graded as 0 (negative), 1: <10%; 2: 10–50%, 3: 51–80%; 4: ≥81%. The expression score (ES) was calculated by the equation ES = P × S.

### 2.4. Oil-Red-O Staining

To examine the amount of fat accumulation, cells were stained with Oil-Red-O (#O0625 Sigma-Aldrich). For Oil-Red-O staining, 5 × 10^4^ cells were seeded into 12-well plates. Briefly, dishes were washed with cold PBS and fixed in 10% formaldehyde for 45 to 60 min. After washing with isopropanol 60%, Oil-Red-O was added and left for 12 min, followed by washing in distilled water [17].

### 2.5. Quantification of Intracellular Fat Accumulation

For further quantification of lipid droplets, Oil-Red-O (#O0625 Sigma-Aldrich) dye was extracted with isopropanol (100%) and quantified by measuring the absorbance at 490 nm (Opsys MR Microplate Reader, Dynex Technologies, Chantilly, VA, USA). Afterward, the cells were stained with Hoechst 33,258 nucleic acid stain (#H3569 ThermoFisher Scientific) to determine cell quantity. Immunofluorescence was measured at 460 nm by Tecan iInfinite F200 (Tecan, Männedorf, Switzerland) [18]. The ratio of Oil-Red-O intensity to cell amount was calculated.

### 2.6. RNA Preparation and Real-Time PCR

Total RNA was isolated (#74106, RNeasy Mini Kit, Qiagen, Hilden, Germany) according to the manufacturer’s instructions. Then, 500 ng of RNA was subjected to reverse transcription (ThermoFisher Scientific, Waltham, MA, USA). cDNA was subjected to real-time PCR (CFX96; Bio-Rad) using the SYBR-Green assay (#M3003, Luna Universal qPCR Master Mix, NEB, Frankfurt, Germany) with gene-specific primers, at a final concentration of 0.2 μM. All samples were analyzed in duplicates and the amounts of expressed mRNA were normalized to *β*-actin mRNA expression. The following primers for mRNA expression were purchased from MWG Eurofins (Ebersberg, Germany):

*β*-actin-F (5′-CTCTTCCAGCCTTCCTTC-3′), *β*-actin-R (5′-AGCACTGTGTTGGCGTAC-3′)

HSF1-F (5′-GCACATTCCATGCCCAAGTAT-3′), HSF1-R (5′-GGCCTCTCGTCTATGCTCC-3′)

CAT-F (5′-ACTGTTGCTGGAGAATCGGG-3′), CAT-R (5′-GAGGGGTACTTTCCTGTGGC-3′)

GPX1-F (5′-CAGTCGGTGTATGCCTTCTCG-3′), GPX1-R (5′-GAGGGACGCCACATTCTCG-3′)

DGAT2-F (5′-GAATGGGAGTGGCAATGCTAT-3′), DGAT2-R (5′-CCTCGAAGATCACCTGCTTGT-3′)

CPT1a-F (5′-CTGCTTTACAGGCGCAAACT-3′), CPT1a-R (5′-TCATGTGCTGGATGGTGTCT-3′)

FASN-F (5′-AAG GAC CTG TCT AGG TTT GAT GC-3′), FASN-R (5′-TGG CTT CAT AGG TGA CTT CCA-3′)

PPARg-F (5′-TGA AGG ATG CAA GGG TTT CT-3′), PPARg-R (5′-CCC AAA CCT GAT GGC ATT AT-3′)

ACACA-F (5′-AGATGGTGGCTGATGTCAAT-3′), ACACA-R (5′-AGGAACGTGTGCAGAATCTT-3′)

FRX-F (5′-TCAGACCGTGAATGAAGACA-3′), FRX-R (5′-GTCTGTTGATCTGGGGTGAG-3′)

HSP70-F (5′-CTGGGTGGGGAGGACTTTG-3′), HSP70-R (5′-GCTTGTTCTGGCTGATGTCC-3′)

Plin2-F (5′-TTGCAGTTGCCAATACCTATGC-3′), Plin2-R (5′-CCAGTCACAGTAGTCGTCACA-3′).

### 2.7. siRNA Transfection

For siRNA transfection, 1 × 10^5^ HepG2 cells were seeded in 12-well plates. After 24 h, 80 pM of the control (#4390846, ThermoFisher Scientific) or HSF1 siRNA (s6951, ThermoFisher Scientific) was mixed with 4 µL of L Lipofectamine RNAiMax (ThermoFisher Scientific, Waltham, MA, USA) in 100 µL of OPTI-MEM and 100 µL of the mixture was applied to the cells for 24 h, followed by medium exchange.

### 2.8. Western Blotting

Preparation of nuclear extracts or total cell lysates was carried out as described before [19,20]. After electrophoresis and wet electroblotting onto PVDF membranes, the following primary antibodies were used for immunodetection at a 1000-fold dilution in 5% (*w*/*v*) non-fat milk powder, 0.05% Tween-20 in TBS (Tris-buffered saline; 50 mM Tris-HCl, pH 7.6, and 150 mM NaCl), including HSF1 (Cell Signaling Technology Cat# 4356, RRID:AB_2120258; 1:1000), beta-actin (Cell Signaling Technology Cat# 4967, RRID:AB_330288, 1:2000), CPT1a (Proteintech Cat# 15184-1-AP, RRID:AB_2084676, 1:10,000). After incubation overnight at 4 °C, blots were exposed to the appropriate horse radish peroxidase-conjugated secondary antibody (Cell Signaling Technology Cat# 7074, RRID:AB_2099233) diluted (1:1000) in blocking buffer and developed using the SuperSignal West Dura Extended Duration Substrate (#34075, ThermoFisher Scientific). Data acquisition was carried out with the Chemidoc-XRS gel documentation system (Bio-Rad, Munich, Germany), using the Quantity One software (Bio-Rad). Beta-actin served as the loading control.

### 2.9. Gel Shift Assays

Cells were treated as described, washed with PBS, lysed in ice-cold EMSA buffer 1 (10 mM HEPES (pH7.9), 10 mM KCl, 0.2 mM EDTA, 1 mM DTT, 0.5 mM PMSF, 10 µ/mL aprotinin, 0.6% TX-100; Phosphatase Inhibitor Cocktail) and centrifuged for 1 min at 21,000× *g*. The supernatant was aspirated and the pellet was washed twice in EMSA buffer 1 (without TX-100). For nuclear protein isolation, the pellet was suspended in ice-cold EMSA buffer 2 (20 mM HEPES (pH7.9), 0.4 M NaCl, 0.2 mM EDTA, 1 mM DTT, 0.5 mM PMSF, 10 µ/mL aprotinin, Phosphatase Inhibitor Cocktail) and incubated on a rocking platform at 4 °C for 30 min. After centrifugation at 21,000× *g* for 15 min, the nuclear protein-containing supernatant was collected. For gel shift assays, 5 µg of the nuclear proteins was incubated with 2 µL of 5× gel shift binding buffer) Promega, Heidelberg, Germany) and 2 µL of γ-P^32^-labeled probe (for Supershift assays, 2 µL of the antibody was added), filled up to a volume of 10 µL with H_2_O and incubated for 30 min at 37 °C. Afterward, native polyacrylamide gel electrophoresis using 5% acrylamide gels in Tris-boric EDTA buffer (pH 7.6) was performed with the labeled samples and the gel was analyzed by autoradiography. Sequences of the probes were as follows: HSE consensus sequence: 5′-CTAGAAGCTTCTAGAAGCTTCTAG-3′ that contained a HSE consensus sequence was used. Supershift analysis confirmed the presence of HSF1 binding to the oligonucleotide by forming a slower migrating supershift complex.

For analysis of the CPT1a promoter: 5′-GAACTCCTGGCTTCAAGCGACTTCCCGCCTCCCG-3′, CPT1a mut: 5′-GAACTCCTGGCTGATAGCGAGCACCCGCCTCCCG-3′Supershift antibodies: two different HSF1 antibodies (Cell Signaling Technology Cat #4356, RRID:AB_2120258, 2µL; Santa Cruz Biotechnology Cat# sc-9144, RRID:AB_2120276, 4 µg).

### 2.10. Statistics

Data represent the mean ± S.D. and were analyzed by Student’s *t*-test; *p*-values < 0.05 were considered statistically significant and were indicated by an asterisk.

## 3. Results

### 3.1. HSF1 Is Abundant in Nuclei of Hepatocytes before and after Bariatric Surgery

To verify the pathophysiological relevance of the observed changes in DNA methylation of HSF1, we first verified its expression in the nuclei in human liver. For this purpose, liver sections of patients before and after bariatric surgery were analyzed by immunohistochemistry. Liver sections were obtained during bariatric surgery (Figure 1a, left panel) and six to twelve months later (Figure 1a, right panel). At this point, patients underwent a control biopsy. Immunohistochemically staining detected HSF1 in the nuclei of hepatocytes in all samples investigated. In line with the observed different methylation of the target sites as a key functional regulation step, there was no significant difference in the percentage of HSF1-stained nuclei or in the intensity of staining in the samples before and after weight loss (Figure 1b).

Furthermore, by using the clinical data of 14 patients, we were able to show that in addition to the expected lowering of the body mass index (BMI), the NAS decreased after bariatric surgery, indicating a reduction in fat accumulation in the post-21bariatric patients (Table 1).

### 3.2. Cultivation with Palmitic and Oleic Acid Induces Fat Accumulation in Hepatic Cell Lines

To investigate the impact of HSF1 on the pathogenesis of NAFLD and to mimic the clinical situation before and after bariatric surgery and/or a dietary treatment, HepG2 and Hep3B cells were cultured with a mixture of palmitic and oleic acid (PA and OA) for 96 h to induce fat accumulation. Afterward, cells were treated with a control medium (fatty acid-free BSA) to induce diminution of intracellular fat accumulation (F-; mimicking the situation after bariatric surgery/dietary weight loss) for 48 h or continually with PA/OA medium (F+, mimicking the situation before bariatric surgery). After treatment with PA/OA medium (F+), Oil-Red-O staining showed increased fat accumulation in HepG2 and Hep3B, which was reversible after cultivation with the control medium (F-) (Figure 2a), mimicking the processes observed in the patients after bariatric surgery. Using another marker for fat accumulation, we analyzed the expression of Plin2 mRNA, which surrounds lipid droplets and correlates with fat accumulation [21]. Therefore, we were able to confirm the observations made by Oil-O staining (Figure 2b).

### 3.3. HSF1 Is Activated during Fat Accumulation in the NAFLD Model

Upon activation, HSF1 shuttles to the nucleus and binds to heat shock elements (HSEs). To evaluate whether PA and OA induce HSF1 shuttling and binding to HSEs in the nucleus of hepatocytes, we conducted gel shift assays in HepG2 and Hep3B cells (Figure 3a). For this purpose, we used nuclear extracts of control, F+, and F− cells. Radiolabeled oligonucleotides harboring a HSE consensus sequence were used (Figure 3a). Supershift experiments with two specific anti-HSF1 antibodies verified the binding of HSF1 to the HSE sequence (arrow in Figure 3a) and led to the formation of a slower migrating supershift complex (Figure 3b). The highest nuclear binding activity of HSF1 to the HSE consensus sequence was observed in F+ cells. Interestingly, culturing F+ cells for 48h with a PA/OA-free diet (F- cells) reduced the activation of HSF1 significantly (Figure 3a).

The highest nuclear binding activity of HSF1 was observed in F+ cells. Interestingly, culturing F+ cells for 48 h with a PA/OA-free diet (F− cells) reduced the activation of HSF1 significantly.

### 3.4. Expression Analysis Revealed Strong Upregulation of CPT1a during Fat Accumulation

We next analyzed the mRNA expression of several genes involved in lipid metabolism, beta-oxidation, autophagy, and heat shock response (Figure 4a,b). Carnitine palmitoyltransferase 1A (CPT1a), which is the rate-limiting enzyme in beta-oxidation, was found to be upregulated due to cultivation in PA/OA-enriched medium (F+). In comparison to F+-cultured cells, CPT1a was downregulated in F− cells (Figure 4a,b). Consequently, we analyzed the mRNA expression of this gene panel in liver tissues of adipose patients undergoing bariatric surgery and in follow-up liver biopsies from the same patients. CPT1a mRNA expression was found to be decreased after weight loss (Figure 4c), confirming the observed effects of the cell culture model.

### 3.5. HSF1 Binds to a Putative Binding Site in the CPT1a Promotor

By using web-based search machines (FASTA), a putative binding site for HSF1 was detected in the CPT1a promoter (Figure 5a). In gel shift experiments with nuclear extracts from F+-treated cells, we were able to document the formation of a complex of oligonucleotide that harbors the predicted binding site of the CPT1a promoter and HSF1. Competition and supershift experiments confirmed the specific binding of HSF1 to the radiolabeled oligonucleotide harboring the CPT1a promotor region in the nuclei of F+ cells, while no binding to oligonucleotides with a mutated CPT1a promoter sequence were detected (Figure 5b).

### 3.6. Knockdown of HSF1 Increases Fat Accumulation in HepG2 and Decreases CPT1a Expression

To analyze the functional role of HSF1 in regulating fat accumulation and in controlling CPT1a expression in this scenario, HSF1 expression was targeted by specific siRNA against HSF1. As already shown, culturing HepG2 cells in PA/OA medium increased fat accumulation. Pre-treatment of the cells with HSF1-specific siRNA showed increased Oil-Red-O staining (Figure 6a), as well as increased Plin2 mRNA expression in PA/OA-cultured cells, compared to the control siRNA-treated cells (Figure 6b,c). CPT1a expression was found to be dependent on HSF1 in PA/OA-treated (F+) cells and (F−)-treated cells, since its expression was decreased after treatment with HSF1 siRNA (Figure 6b). Remarkably, in F− cells, this effect was less pronounced (Figure 6b,c) due to the lower activity of HSF1 (Figure 3a).

### 3.7. Activation of HSF1 by Celastrol Diminishes Fat Accumulation in HepG2 Cells

Celastrol is known to activate HSF1 by causing the dissociation of HSF1 from the Hsp70-HSF1 complex and consequent shuttling of HSF1 to the nucleus and binding to DNA. Since our data suggested that HSF1 negatively regulates fat accumulation, we examined whether celastrol administration to HepG2 cells reduces intracellular fat accumulation. Celastrol was applied to cells cultured with a control medium, as well as to cells treated with PA/OA medium, for 96 h in different concentrations (0.025, 0.05, 0.1, 0.2 μM). Intracellular fat accumulation was measured by examining Oil-Red-O intensity. In F+ cells, celastrol treatment at 0.1 μM significantly reduced PA/OA medium-induced fat accumulation (80%) compared to cells treated with mock (DMSO) (Figure 7a). This effect was abrogated if HSF1 siRNA was administered before celastrol treatment (Figure 7b).

## 4. Discussion

We were able to show that HSF1 binding sites are strongly enriched in methylation assays (>400-fold) of liver tissue of NAFLD patients in the process of liver remodeling, after significant weight loss following bariatric surgery [10]. This pure observational finding indicated a potential role of HSF1 in NAFLD disease progression, but up to now, functional data that confirm this assumption have been missing.

By using a well-defined in vitro model that mimicked the situation of patients before bariatric surgery (F+ cells), we demonstrated that HSF1 activity is upregulated during lipid accumulation. When these cells were set on a PA/OA-free diet (F− cells), decreasing amounts of intracellular lipid droplets were accompanied by declining HSF1 activity. The functional outcome of HSF1 activation in NAFLD is controversial [10,11,12,13,14,22,23,24,25,26]. Some reports indicate that HSF1 activation leads to the progression of NAFLD and, in the end, to hepatocellular carcinoma by stimulating lipid biosynthesis [23]. However, most of the reports state that HSF1 dysfunction plays an important role in the development of NAFLD, by showing a negative correlation between HSF1 expression/activation and progression from NAFLD to fibrosis in liver and adipose tissues [12,13,14,22,24,26]. In this context, di Naso et al. analyzed NAFLD liver biopsies and observed decreasing rates of HSF1-positive cells, as NAFLD progresses from steatosis to steatohepatitis to steatohepatitis plus fibrosis [27]. In line with this, pharmacological activation of HSF1 by celastrol or SYSU-3d ameliorates NAFLD in cellular and animal models. Enhanced HSF1 activity by these pharmacological agents was accompanied by the activation of the PGC-1α pathway, which is essential for cellular energy metabolism [14,24]. We were able to confirm these effects of HSF1 activation on fat accumulation in our model system. Activation of HSF1 by celastrol had beneficial effects, while siRNA-mediated knockdown of HSF1 increased fat accumulation in the cellular model. These data support the observation that HSF1 activation is one of the mechanisms used by the body to counteract steatosis. In line with this, the observed differential methylation status of HSF1 binding sites in liver tissue of NAFLD patients in the process of liver remodeling [10] must be interpreted as an indirect marker for the reduction in fat through therapeutic measurements. The target genes of HSF1 in the steps of NAFLD disease progression involve chaperones, such as heat shock proteins [13,26], PPARgamma coactivator-1alpha [14,24], and the CaM-Akt pathway [22]. Interestingly, we were not able to demonstrate the regulation of these pathways in our model. After analyzing the expression of a panel of lipid metabolism-associated genes in PA/OA medium (F+) and control medium (C/F−)-treated cells, carnitine palmitoyltransferase 1A was identified as a regulated gene. CPT1a is inhibited by glucose metabolism and the first intermediate of lipogenesis, malonyl-CoA, and makes CPT1a the rate-limiting step in fatty-acid beta-oxidation. The idea of enhancing hepatic fatty-acid oxidation in order to improve obese metabolic phenotypes is discussed in the literature. In this context, even a moderate activation of CPT1a and subsequent increased fatty-acid oxidation in the liver resulted in a decreased hepatic triacylglyceride content in obese rodents [28]. CPT1a-overexpressing mice were protected against obesity-induced weight gain, obesity-induced insulin resistance, and hepatic steatosis [29,30]. Despite the plethora of reports showing upregulation in cellular and animal models, the exact upstream events that lead to increased expression of CPT1a are unclear [31]. A recent report indicated that the expression of the gene is regulated by methylation and by binding of factors such as C/EBPbeta, PPARalpha, PGC1alpha, and BAF60a to the gene [32]. Interestingly, no direct regulation of CPT1a by HSF1 has been reported. We were able to show the binding of HSF1 to the putative binding site in the CPT1a promoter by gel shift assays. Furthermore, by specific siRNA-mediated knockdown of HSF1 expression, we were able to establish the pivotal role of this transcription factor in the observed upregulation in our model. These in vitro data were confirmed translationally by analysis of the liver biopsies of patients 5–9 months after bariatric surgery, compared to their pre-surgery status. It is tempting to speculate that when using genome-wide analysis for chromatin accessibility, more signatures and target genes could be detected. However, the current manuscript focused on deciphering regulatory processes of the proposed role of HSF1 in NAFLD disease progression in a simplified model system and focused on one transcription factor. Furthermore, we used established methods for analyzing these processes with a limited number of potential target genes and were able to describe a potential pathway for interventional pharmacological therapy.

## 5. Conclusions

In conclusion, we were able to confirm the functional HSF1-CPT1a signaling pathway at different stages of NAFLD and describe this pathway as a potential pharmacological target for the clinical management of metabolic fatty liver diseases.

## Figures and Tables

**Figure 1 cells-11-03504-f001:**
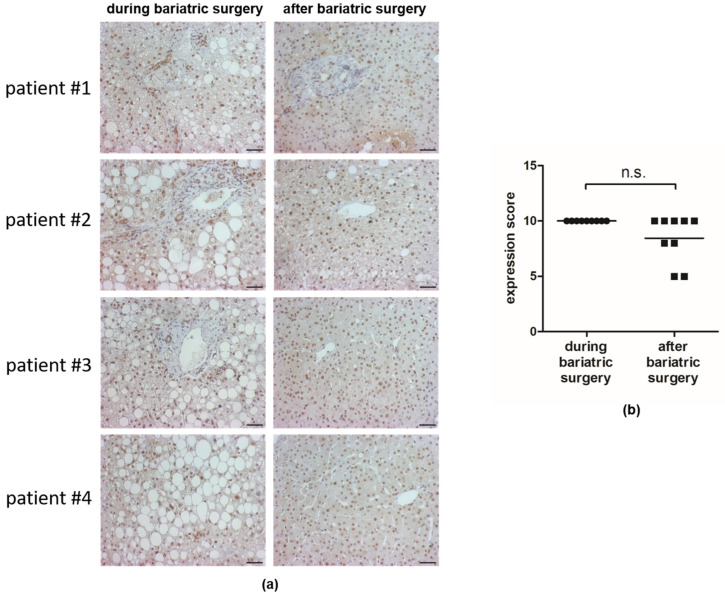
HSF1 staining of hepatocytes before and after bariatric surgery. Representative images of immunohistochemical staining of HSF1 in liver tissue obtained during bariatric surgery ((**a**), left panel) and 6 to 12 months after bariatric surgery ((**a**), right panel) (scale bar 100 μm) are shown. The expression score (ES = P × S) ± mean of HSF1 nuclear staining of 9 liver tissues is depicted (**b**).

**Figure 2 cells-11-03504-f002:**
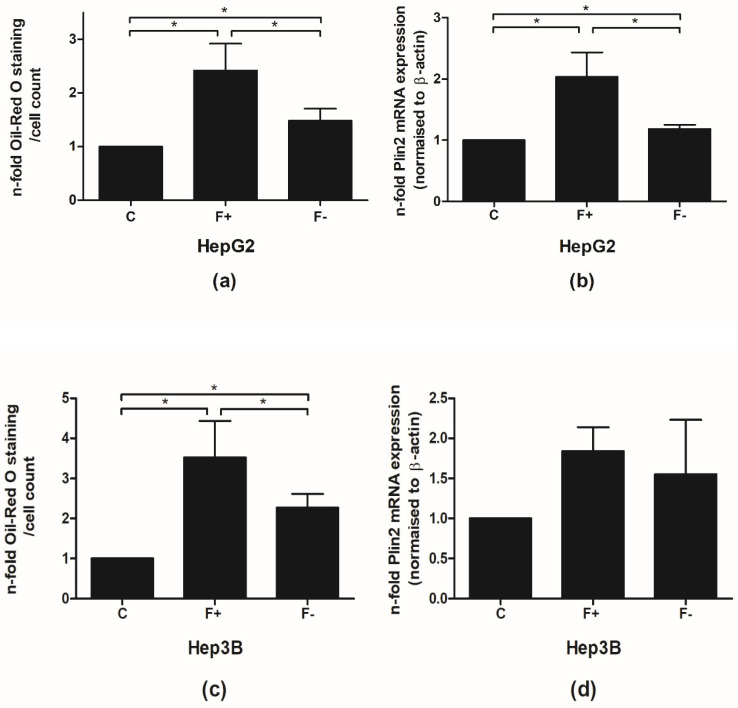
Cultivation with palmitic and oleic acid induces fat accumulation in hepatic cell lines. HepG2 and Hep3B cells were cultured with control medium (C) or medium supplemented with a mixture of palmitic and oleic acid (PA/OA medium) for 96 h to induce fat accumulation. Afterward, cells were treated with a control medium to induce diminution of intracellular fat accumulation (F-) or continually with PA/OA medium (F+), mimicking the situation before bariatric surgery. Oil-Red-O staining (**a**,**c**) and expression of Plin2 mRNA (**b**,**d**) of HepG2 and Hep3B cells are shown. The figure depicts the mean values of five independent experiments performed in duplicates ± S.D. * *p*-values < 0.05.

**Figure 3 cells-11-03504-f003:**
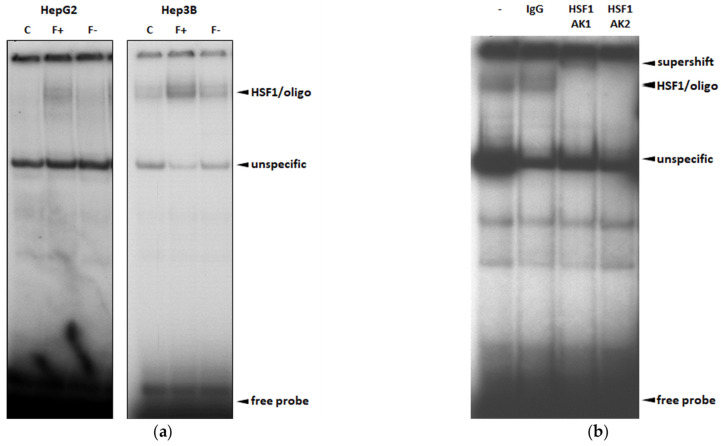
HSF1 is activated during fat accumulation in the NAFLD model. To evaluate whether PA and OA induce nuclear HSF1 binding to HSEs, we conducted gel shift assays with oligonucleotides harboring a consensus HSE in HepG2 and Hep3B cells (**a**). For supershift experiments, nuclear extracts of Hep3B cells were treated with two antibodies directed against different sites of HSF1 (**b**). Representative results of four independent experiments are shown.

**Figure 4 cells-11-03504-f004:**
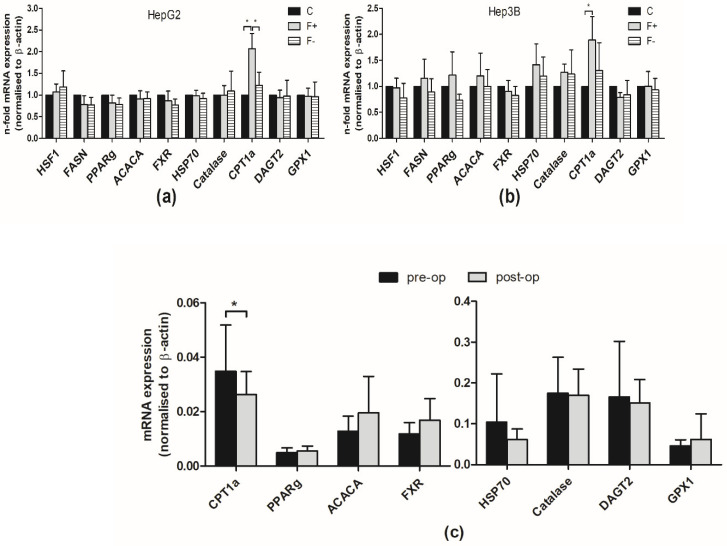
CPT1a is upregulated during fat accumulation. HepG2 and Hep3B cells were cultured as already described. Isolated total mRNA was subjected to reverse transcription and expression of indicated genes was analyzed by real-time PCR (**a**,**b**). For normalization, β-actin expression was analyzed. The figure displays the mean values of three independent experiments performed in duplicates ± S.D. * *p*-values < 0.05. Total RNA from liver tissues of 14 adipose patients undergoing bariatric surgery (pre-op) and of follow-up (post-op) liver biopsies from the same patients were isolated, reverse transcribed, and analyzed by qPCR for indicated gene expression (**c**).

**Figure 5 cells-11-03504-f005:**
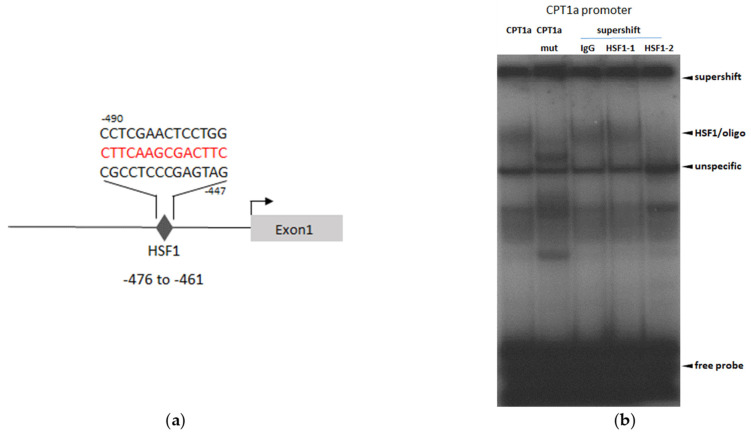
HSF1 binds to a putative HSF1 binding site in the CPT1a promoter. Sequence analysis identified a putative binding site for HSF1 (labeled in red) in the CPT1a promoter (**a**). Nuclear extracts from F+ cells were analyzed by gel shift assays with oligonucleotides that spanned the putative HSF1 binding site (lane 1,3,4,5) or a mutated HSF1 binding site (lane 2). Additionally, supershift experiments with two HSF1 specific antibodies, as well as an IgG antibody as the control, were performed (**b**). Representative results of four independent experiments in Hep3B cells are shown.

**Figure 6 cells-11-03504-f006:**
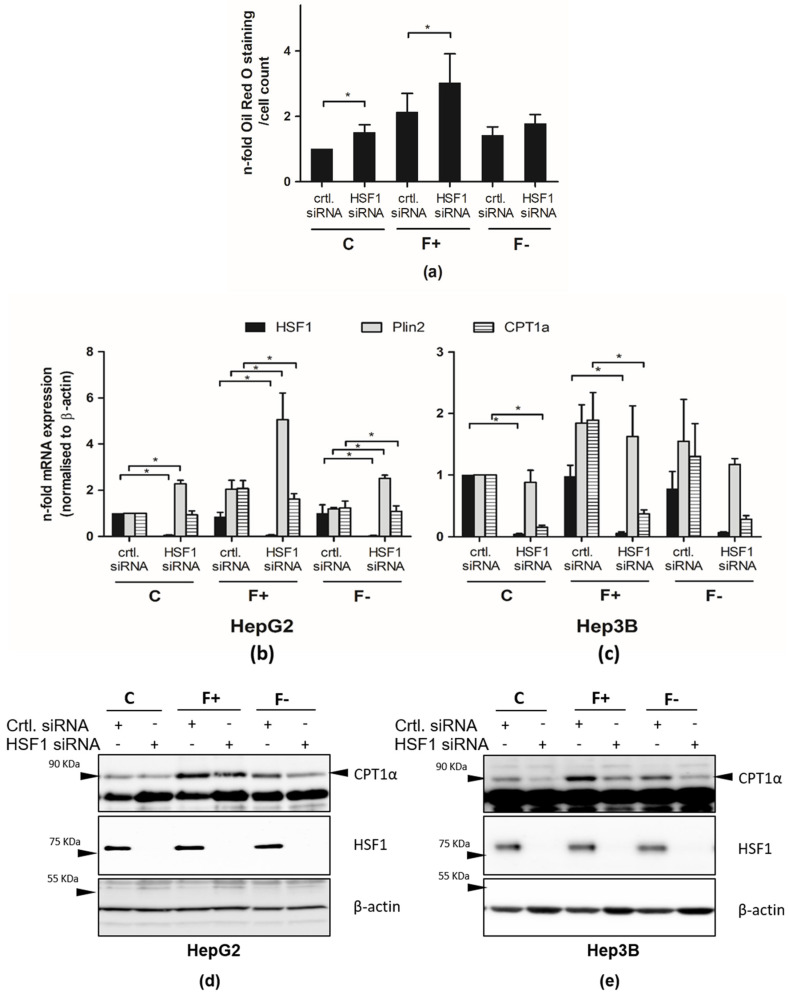
Knockdown of HSF1 increases fat accumulation in HepG2 and Hep3b and decreases CPT1a expression. HepG2 and Hep3B cells were transfected with the indicated siRNA and cultured with control (C) or PA/OA medium for 96 h to induce fat accumulation. Afterward, PA/OA medium-cultured cells were treated with a control medium (F-) or continually with PA/OA medium (F+). Oil-Red-O staining (**a**) of HepG2, mRNA expression of indicated genes in HepG2 and Hep3b cells (**b**,**c**) and protein expression of the indicated genes (**d**,**e**) are shown. For normalization, β-actin expression was analyzed. The figure depicts the mean values of three independent experiments performed in duplicates ± S.D. * *p*-values < 0.05. For protein expression, representative results of 4 independent experiments are shown.

**Figure 7 cells-11-03504-f007:**
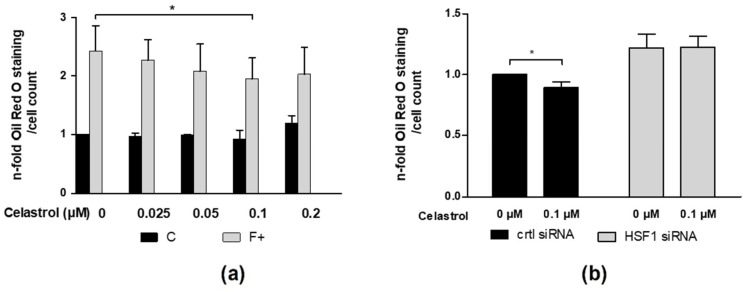
Activation of HSF1 by celastrol diminishes fat accumulation in HepG2 cells. HepG2 cells were stimulated with indicated concentrations of celastrol and cultured in control or PA/OA medium for 96 h. Intracellular fat accumulation was measured by Oil-Red-O staining (**a**). HepG2 cells were transfected with HSF1 or control siRNA. After 24 h, cells were treated with 0.1 µM celastrol and cultured in control or PA/OA medium for 96 h. Fat accumulation was measured by Oil-Red-O staining (**b**). The figures depicts the mean values of three independent experiments performed in duplicates ± S.D. * *p*-values < 0.05.

**Table 1 cells-11-03504-t001:** Patient characteristics of 14 adipose patients undergoing bariatric surgery (pre-op) and after follow-up (post-op).

Parameter	Pre-Op	Post-Op
Number of patients	14	14
Age in years (mean ± SD)	46 ± 12.1	46 ± 12.1
Female/male (number)	12/2	12/2
BMI (kg/m^2^) (mean ± SD)	51.8 ± 9.8	38.1 ± 9.1
NAS (mean ± SD)	3.1 ± 2.3	1 ± 1.2

## Data Availability

Not applicable.

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
