# Peer review of "The HSF1-CPT1a Pathway Is Differentially Regulated in NAFLD Progression"

_cells, 2022, doi:10.3390/cells11213504_

Round 1
Reviewer 1 Report
This is an interesting manuscript on the role of HSF1 in NAFLD patients using a precious set of samples (pre- and post bariatric surgery). In addition, the authors use an established in vitro model to show that HSF1 is upregulated during fat accumulation. Upregulation of HSF1 is accompanied by CTP1a upregulation while knock-down of HSF1 by siRNA results in CTP1a downregulation. Since downregulation of HSF1 results in reduced fat accumulation in vitro, the authors speculate that HSF1 is a protective mechanism during NAFLD progression counteracting excessive fat accumulation and progression to fibrosis/cirrhosis.
Although the manuscript is well written and the experiments soundly performed, there are few concerns that should be considered to further improve the manuscript:
1. The authors should tone down on the previously performed array-based DNA methylation experiments (abstract, introduction, discussion) and highlight the fact that these experiments were published in a separate paper a couple of years ago.
2. Figure 1 should be extended if possible. Additional stainings of fat accumulation (e.g. oil o red) could be provided in a set of pre-and post-OP samples including a quantification that confirms the representative impression of fat reduction post bariatric surgery.
3. Please add scale bars to the images.
4. Figure 4: Is there an antibody available allowing IHC or IF for CPT1a/ (and or Plin2) in pre-and post bariatric surgery samples to confirm the mRNA results and in vitro findings? Again, quantification of a set of samples would be desirable.
5. Downregulation of HSF1 is accompanied by increased fat accumulation and differential regulation of Plin2 and CPT1a (Figure 6a+b) in HepG2. Can the authors provide data from a second cell line to underline the relevance of their findings.
6. Obvious shortcomings (that are beyond the scope of the current manuscript) should be briefly discussed (e.g. CHIP, ATACseq etc.)
Reviewer 2 Report
The authors demonstrated that the knockdown of HSF1 can affect the expression level of CTP1a in liver cell lines, and used gel shift experiments to demonstrate that HSF1 can bind to the promoter region sequence of CTP1. At the same time, the authors demonstrated that the knockdown of HSF1 can reduce the effect of Celastrol for blocking intracellular fat accumulation. In addition, the authors also analyzed the expression levels of HSF1 in clinical samples. The authors concluded that changes in HSF1 transcriptional activity may have led to clinical improvement in patients. The manuscript has the following major flaws:
1. In Figure 1, the authors analyzed the expression of HSF1 in liver sections from patients during and after bariatric surgery by immunohistochemistry. Two IHC images are shown. The authors concluded that HSF1 expression in liver nuclei did not change significantly before and after surgery. Here the authors may be trying to demonstrate that it is not a change in the level of HSF1, but a change in the activity of HSF1, that leads to postoperative improvement in patients. More clinical samples need to be shown if the authors attempt to use clinical samples to claim in the manuscript. Plus, there may be many factors that contribute to the improvement of NAFLD from bariatric surgery. Decreased nutrient intake may be an important factor, but it is not known whether HSF1, as highlighted by the authors, is downstream of decreased nutrient intake. Therefore, it is recommended that the author move the clinical information of figure 1 to the supplementary material or the last figure to avoid logical confusion.
2. In Figure 2, the authors treated in vitro HepG2 or Hep3B cells with palmitic and oleic acid. The lipid accumulation in F+ cells was verified by oil red staining and Plin2 mRNA levels. In Figure 3, Gel shift experiments were performed using F+/F- cells in Figure 2. In the manuscript, the authors mentioned that probes called HSEs seem to be added to the system to detect the binding capacity of HSF1 and HSEs. In Figure 3A, it appears that high migratory signals are seen in F+ cells. As a rule of thumb, Figure 3A-B should be autoradiography images, but the legend does not indicate what those images are, and the labeling of HSF1 on the way is misleading, because the signal indicated by the arrow is likely to be the autoradiographic signal of HSEs, but not the protein signal of HSF1. In Figure 3B, the authors used two antibodies to verify the existence of migration, but there is the same problem as in Figure 3A, that is, it is not known whether the signal is HSF1 or HSEs. Also, this part is not described in detail in the Methods section.
3. In Figure 4, the authors examined the expression of several different mRNAs associated with corresponding biological processes. The mRNA expression of CPT1a was found to be significantly elevated in F+. And the expression of CPT1a in surgical samples before and after surgery was detected. The authors proposed that CPT1a is significantly downregulated after surgery. Here the authors should examine the expression of several different mRNAs associated with the corresponding biological process in the pre- and post-surgery samples, and only if similar trends are found in the surgical and cellular samples can the authors' claims in the manuscript be clarified.
4. In Figure 5, the authors used software to predict the binding site of HSF1 in the CPT1 promoter. The Gel shift assay was used to detect the mobility changes of WT and mutant CPT1 promoter DNA sequences as probes. The mutated CPT1 promoter DNA sequence was found to exhibit poorer binding compared to WT. However, there is a problem similar to Figure 3 in this result, that is, the labeling is not clear. The arrow in the figure indicates not HSF1 protein, but the signal of DNA fragment autoradiography; the legend indicates that this figure is called EMSA experiment, which has the same meaning as gel shift in Figure 3, this is inconsistent with the previous term.
5. In Figure 6, the authors demonstrated that HSF1 increases fat accumulation in HepG2 and decreases CPT1a expression through Knockdown experiments. In the legend, the authors mention that the same experiment was performed in Hep3B but the results did not appear, and there is no description of this part in the figure nor in the manuscript. The results of this part should be included. If the two cell lines have inconsistent results, possible reasons should be discussed in the Discussion. Additionally, rescue experiments overexpressing HSF1 should be performed to demonstrate the specificity of the effect of HSF1 knockdown.
6. In Figure 7, the authors activated HSF1 by adding the drug Celastrol, and found that the accumulation of fat was significantly reduced in HepG2 cells in the drug group. Here the authors shown that the knockdown of HSF1 cells does not cause a significant reduction in fat accumulation. Here, rescue experiments of overexpressing HSF1 should be introduced to demonstrate the specificity of the effect of the knockdown of HSF1.
The following minor issues remain in the manuscript:
In the materials and methods, the author mentioned western blot, but in the manuscript and pictures, we did not see any western blot results. The authors need to carefully check.
Overall, the manuscript is novel in concept and the results are real. It is recommended to be published after sufficient revision.
Round 2
Reviewer 1 Report
The manuscript has improved follwing revision. However, I cannot identify a brief discussion of the shortcomings (# 6) as requested. The authors should include this in the discussion.
Author Response
Response to Reviewers
We thank all reviewers for time, effort and constructive criticism, pursuing to improve the manuscript. Especially we are delighted about the overall positive evaluation of our revised work.
Please see below the point by point responses.
Reviewer #1:
Obvious shortcomings (that are beyond the scope of the current manuscript) should be briefly discussed (e.g. CHIP, ATACseq etc.)
Response by the authors:
We are sorry that we did not discuss this important point and included it in the revised discussion section.
Reviewer 2 Report
The reviewers hold a positive attitude for authors’ innovative hypothesis and verification for the phenomenon that HSF1 protein did not change before and after surgery may be due to changes in its regulatory functions. Thanks to the authors for their knowledge and the revision and responses. But the author's responses still have the following problems:
Question 1. The authors provide a new table of clinical information that describes some of the clinical baseline and NAS information. However, there is still a lack of information on HSF1 IHC for the additional patient samples presented in Figure 1 to clarify the authors' point that expression levels did not change before and after surgery. In other words, a single patient IHC is not quantitative enough to express the author's point. It is recommended to supplement more IHC IMAGES, and the results of quantitative analysis of the images. In addition, the statistical expression in the new table supplemented by the author is not very scientific. The table does not indicate whether the statistic is the mean or the median, and does not indicate the positive and negative error range.
Question 2. We think the author did not respond well to the question. The reasons are as follows: 1) As far as Figure 3 and Figure 5 are concerned, the position indicated by the HSF1 arrows marked by the author does not necessarily exist HSF1 protein, and regardless of whether the author did not use biochemical methods to verify that the band is really HSF1 protein. The principle of motion of molecules in an electric field, we have good reason to suspect that HSF1 and the DNA probe do not exist in one location. Because the movement of the probe DNA molecules is only hindered, it does not mean that the probe DNA molecules must move in a bound state with the protein in the gel. Therefore the position indicated by the arrow does not necessarily have a protein. 2) It is obvious that some authors’ arrows have indicated more than one band in the positions, but the authors only marked one arrow, so how to interpret other bands, whether those extra bands contain probe molecules or probe molecules + HSF1 protein, or are they just HSF1 protein?
Question 3. For this question, it seems that the authors do have difficulty in obtaining mRNA data from more clinical samples.
Question 4. No comments.
Question 5. Sorry, reviewer 2 can't see how the author responded to reviewer 1 here.
Question 6. There are not enough responses from the author here. The question of rescue experiment remains in question 5. Here reviewer will explain why the rescue experiment is needed. When siRNA molecules act on cells and produces a biological effect, we expect that its action process is that the mRNA of this protein is knocked down, resulting in a decrease in protein, which in turn leads to a biological effect by the protein. Not to mention, the authors did not prove whether the protein was actually reduced in this experiment. Just to discuss the possibility that the biological process studied by the author is only sensitive to the knockdown of HSF1 process, that is to say, knockdown of HSF1 is only indirectly related to the subsequent biological consequence. For example, it is possible that the mRNA of HSF1 itself is involved in the biological process of the authors’ study, rather than the protein involved. As another example, it may be that a certain level of decreasing in non-specific mRNA may lead to the biological process the authors studied. Here, the control non-specific siRNA is not enough, because it is likely that the control non-specific siRNA is not enough to cause any decrease in any mRNA, and the biological process studied by the author is likely to occur when there is a significant decrease in a non-specific mRNA. Here, unless the authors demonstrate that knocking down another mRNA does not cause the phenotype (Any mRNA, whatever, actin, GAPDH or other), if not, rescue experiments is needed.
Minor issues. Thanks to the authors for removing the western blot method.
Overall, the authors did not respond well to the questions raised by the reviewers. However, in view of the innovativeness and real scientific attitude of this paper, the reviewers suggest that the paper can be published after conducting additional experiments and a deep revision.
Round 3
Reviewer 2 Report
After two rounds of revisions, the manuscript appears to be much improved. The reviewers suggested that the manuscript could be published without further revision.
Although the reviewers still have different opinions on questions 2 and 5, especially question 5.
Regarding question 5, the reviewers agree with the authors that the phenotype of some molecules cannot be rescued by simple overexpression. This can be analogous to removing one part in a car engine, and it will not rescue the engine by putting more parts in the engine, more parts will make the engine more damaged. However, the reviewers believe that by reasonably mimicking the expression of molecules in vivo, such as endogenous promoters, the expression of the protein will be more in line with physiological conditions. And even if the further downstream cannot be rescued, the molecular phenotype of direct downstream should also be rescued, such as molecules that interact directly.